# Compaction of a Polymeric Membrane in Ultra-Low-Pressure Water Filtration

**DOI:** 10.3390/polym14163254

**Published:** 2022-08-10

**Authors:** Muhammad Roil Bilad, Siti Rahma Junaeda, Yusran Khery, Baiq Asma Nufida, Norazanita Shamsuddin, Anwar Usman, Violet Violet

**Affiliations:** 1Faculty of Applied Science and Education, Universitas Pendidikan Mandalika, Jl. Pemuda No. 59A, Mataram 83126, Indonesia; 2Faculty of Integrated Technologies, Universiti Brunei Darussalam, Gadong BE1410, Brunei; 3Department of Chemistry, Faculty of Science, Universiti Brunei Darussalam, Jalan Tungku Link, Bandar Seri Begawan BE1410, Brunei; 4Faculty of Forestry, Lambung Mangkurat University, Jl. A. Yani KM. 36, Banjarbaru 70714, Indonesia

**Keywords:** membrane compaction, gravity-driven membrane filtration, water and wastewater treatment, sustainable engineering

## Abstract

Applications of ultra-low-pressure filtration systems are increasing as they offer enhanced sustainability due to lower energy input, almost no use of chemicals, and minimum operational expenditure. In many cases, they operate as a decentralized system using a gravity-driven membrane (GDM) filtration process. These applications are relatively new; hence, the fundamental knowledge of the process is still limited. In this study, we investigated the phenomenon of polymeric membrane compaction under an ultra-low-pressure system. The compaction phenomenon is well-recognized in the traditional pressure-driven system operating at high transmembrane pressures (Δ*P*s > 200 kPa), but it is less documented in ultra-low-pressure systems (Δ*P* < 10 kPa). A simple GDM filtration setup operated under a constant-pressure system was employed to investigate the compaction phenomena in a polymeric hollow fiber membrane for clean water filtration. Firstly, a short-term pressure stepping test was performed to investigate the occurrence of instantaneous compaction in the Δ*P* range of 1–10 kPa. The slow compaction was later investigated. Finally, the compaction dynamic was assessed under alternating high and low Δ*P* and relaxation in between the filtrations. The findings demonstrated the prominence of membrane compaction, as shown by the decreasing trend in clean water permeability at higher Δ*P*s (i.e., 3240 and 2401 L m^−2^ h^−1^ bar^−1^ at Δ*P*s of 1 and 10 kPa, respectively). We also found that the intrinsic permeability of the applied polymeric membrane was significantly higher than the apparent one (4351 vs. 2401 L m^−2^ h^−1^ bar^−1^), demonstrating >50% loss due to compaction. The compaction was mainly instantaneous, which occurred when the Δ*P* was changed, whereas only minor changes in permeability occurred over time when operating at a constant Δ*P*. The compaction was highly reversible and could be restored (i.e., decompaction) through relaxation by temporarily stopping the filtration. A small fraction of irreversible compaction could be detected by operating alternating filtrations under Δ*P*s of 1 and 10 kPa. The overall findings are essential to support emerging GDM filtration applications, in which membrane compaction has been ignored and confounded with membrane fouling. The role of compaction is more prominent for high-flux GDM filtration systems treating less-fouling-prone feed (i.e., rainwater, river water) and involving membrane cleaning (i.e., relaxation) in which both reversible and irreversible compaction occurred simultaneously.

## 1. Introduction

The gravity-driven membrane (GDM) system is one of the current configurations used for the sustainable filtration of water and wastewater. The system has been developed for decentralized systems and implemented in some full-scale submerged membrane bioreactors (MBRs). It applies to a wide range of water and wastewater treatment applications [1]. The system operates at extremely low pressures (Δ*P* < 10 kPa) compared with the traditional microfiltration (MF) or even ultrafiltration (UF) processes (Δ*P* of 100–500 kPa); hence, it requires much lower energy consumption [2,3]. Stabilization of the flow in the range of 2–20 L/m^2^h was achieved in a laboratory-scale test without backwashing or routine cleaning [4,5], producing a maintenance- and cleaning-free system. Thus, GDM may play a significant role as an energy-saving strategy in advanced wastewater and water treatment.

The progress of GDM developments has been extensively reported [1]. However, the limitation of GDMs is the relatively low stable flux over long-term operation. It is attributed to the formation of a fouling layer on the membrane surface. Due to the dead-end filtration system, organic debris and biological organisms in the feed are permitted to build on the membrane surface and form a (bio)cake layer, increasing the permeation resistance. Membrane fouling is overwhelmingly acknowledged in GDMs. Attempts have been made to manipulate the (bio)cake layer to enhance the system throughputs [6,7,8]. Yet another phenomenon, compaction, also takes place. Compaction, in turn, plays an important role, especially under ultra-low-pressure filtration, as demonstrated in this study. Process engineers often overlook the occurrences of compaction by considering it simply as membrane fouling.

Membrane compaction has long been recognized to occur in polymeric membranes under compression. The mechanical stability of porous polymeric membranes depends on their structures, and mechanical characteristics compaction preferentially occur in the bulk layer where large pores and macrovoids are situated [9]. A sponge-like structure was less affected by compaction than a structure with macrovoids [10]. These findings suggest a close relationship between the fabrication parameter and the compaction phenomena. In a pressure-driven filtration system, the applied Δ*P* exerts the compressive forces required to deform the membrane structure, which physically alter the membrane properties [9] and filtration performance. Due to compressive strain, the physical deformation may lower the porosity, constrict the pores, and densify the polymer matrix. The polymer matrix can exhibit instantaneous or slow deformation [11] as a result of elastic, plastic, and viscoelastic strains (i.e., plasticization, swelling, and creep) [11,12,13,14]. When the applied Δ*P* is released, the thickness instantaneously may increase, returning to the precompression state. However, many polymeric materials exhibit irreversible deformation, as they do not completely restore to the initial condition upon the release of the compressive force [15]. In this regard, the two different compaction behaviors are referred to as reversible and irreversible compactions. The latter results in irreversible flux degeneration even under relatively low Δ*P*. For example, Tessaro and Jonsson [16] discovered that after temporarily exposing a polysulfone UF membrane to a constant Δ*P* of 180 kPa, the filtration capacity of the membrane significantly decreased. Kallioinen et al. [17] also discovered that a UF membrane under compaction at 300 kPa lost its filtration capacity. In more recent reports on ultra-low-pressure filtration systems, the clean water permeability of a hollow fiber membrane decreased by >43% when Δ*P* increased from 2.2 to 10.0 kPa [18]. More importantly, a flat sheet membrane lost its clean water permeability by >86% due to compaction when Δ*P* increased from 2.5 to 19.0 kPa [19]. All these reported findings demonstrate the significance of membrane compaction in ultra-low-pressure filtration systems.

Along with the above-mentioned flux reduction, membrane compaction can affect separation efficiency and enhance the retention of molecules of the same magnitude as or smaller than the membrane cutoff value, as membrane compaction may lower pore size or alter pore geometry. However, Tarnawski and Jelen [20] discovered that while the permeability of a polysulfone UF membrane significantly decreased due to compaction in the pressure range of 0–10 bar, the membrane’s selectivity remained unchanged. Although slow compaction gradually occurs, a lack of adequate assessment of the effect of compaction on membrane performance can significantly skew the results of laboratory-scale studies [21]. In addition to decreasing the flux, compaction influences solute rejection. It may shrink pore size or distort pore geometry and eventually increase solute rejection [22,23]. Despite the increasing amount of research on ultra-low-pressure membrane processes, the fundamental aspect of membrane compaction remains a research challenge.

This study investigated membrane compaction and its effects on ultra-low pressure filtration systems handling no-fouling feed (clean water). Numerous aspects of compaction were examined, including the influence of Δ*P*, reversibility, and dynamics. A better understanding of the membrane compaction behavior can further open new research avenues to establish sustainable GDM processes.

## 2. Methodology

### 2.1. Material

Clean water was obtained by filtration of tap water using a household reverse-osmosis unit, resulting in a total dissolved solute of <20 ppm. The clean water was used in all filtration experiments, utilizing a commercial U-shaped hollow fiber membrane (Xiamen Aqusta Water Technology Co., Ltd., Fujian, China) made from polyacrylonitrile with a nominal pore size of 0.01 µm and a surface area of 0.242 m^2^. U-shaped means the HF is bent inside the module, with both ends seen in the module, which is the typical packing configuration for dead-end filtration. The module had a long cylindrical shape with a base diameter of 4.6 cm and height of 22.5 cm, resulting in a module packing density of 647 m^2^/m^3^.

### 2.2. Filtration Setup

Figure 1 illustrates the ultra-low-pressure filtration setup used for the filtration test in this study. Filtration was driven by the hydrostatic pressure from the elevated feed liquid level above the membrane. The feed was raised above the membrane, which was placed on the base of the polyvinyl chloride cylinder tank with an inner diameter of 10.12 cm. The water level was regulated by opening and closing the ball valves placed every 10 cm up to 100 cm above the baseline (in the middle point of the membrane module), 11 cm from the base of the U-shaped module. This arrangement allowed the hydrostatic pressure variation of 1.0 to 10.0 kPa (0.01–0.1 bar) with an interval of 1 kPa. The feed was recirculated at a rate of 1.5 L/min. The feed flow direction was from the outside to the lumen of the fiber and through the lumen toward the end of the module to allow for permeate collection.

The feed water was recirculated using a liquid pump at a constant rate of 1.5 L/min, and the excess flow was returned to the feed store to maintain the liquid level during filtration. The applied recirculation rate was sufficient to maintain a stable level of the liquid. Fluctuation of the liquid level was only observed for filtration under transmembrane pressures (Δ*P*) below 3 kPa. Before usage, the membrane was wetted by washing with a detergent solution at 60 °C for 15 min and with running tap water for 15 min, followed by immersion in distilled water overnight for pore wetting and activation. The membrane was then kept wet over the entire study to prevent pore deactivation due to drying.

### 2.3. Filtration Test

#### 2.3.1. Flux and Permeability

The permeate flux (J, L m^−2^ h^−1^) and permeability (L, L m^−2^ h^−1^ bar^−1^) were calculated using Equations (1) and (2):(1)$$J=\frac{V}{t\;A}\;$$
(2)$$L=\frac{J}{\Delta P}\;$$
where V is the volume of the permeate collected (*L*), A is the membrane surface area (m^2^), t is the time taken for the collected permeate (h), and ΔP is the applied transmembrane pressure (bar).

#### 2.3.2. Instantaneous and Long-Term Compaction

For instantaneous compaction, filtration was performed for only 30 min under a Δ*P* of 1.0–10.0 kPa, corresponding to 10–100 cm liquid levels. The Δ*P*s were set ascending from 1.0 to 10 kPa at an interval of 1 kPa. The tests were performed in duplicate to detect the permeability as a function of pressure and to estimate the intrinsic permeability without membrane compaction. In this test, we observed a rapid decline in permeability at the beginning of the filtration. A relaxation of 15 min was performed in between the liquid filtration. The permeate tube was closed during the relaxation, and no water permeation was allowed.

Long-term filtration was performed for 8 h continuously under Δ*P*s of 1.0, 6.0, and 10.0 kPa to observe any degeneration in water permeability over time, which could be attributed to the slow compaction of the polyacrylonitrile hollow fiber membrane. Permeability data were taken every 30 min.

#### 2.3.3. Pressure Relaxation and Compaction Dynamics

The effect of abrupt changes in the applied pressure on the membrane compaction was evaluated using multiple filtrations involving relaxations. Six clean water filtrations were performed for one hour of sandwiching and one hour of relaxation. The permeate line was closed during the relaxation to avoid any permeation for one hour in duplicate. This filtration test allowed relaxation to restore permeability by decompaction of the membrane.

The final assessment of the membrane compaction was performed to observe its dynamics under variable pressures. For four cycles, clean water filtrations were alternatingly conducted under Δ*P*s of 1.0 and 10.0 kPa. Each filtration was performed for one hour and in duplicate. This test was conducted to observe the compaction and decompaction dynamics due to the change in Δ*P*.

## 3. Results and Discussion

### 3.1. Instantaneous Compaction

Figure 2 shows the evolution of permeability during the ascending followed by descending Δ*P*. Because no membrane fouling occurred during the filtration of demineralized water, the changes in permeability under various Δ*P*s were fully associated with the change in the membrane resistance in response to the changes in the applied Δ*P*. The figure clearly shows the occurrence of membrane compaction from the decreasing trend in the permeability at higher Δ*P*s. Without any compaction, the clean water permeability did not change. In the first pressure stepping filtration test, the initial clean water permeability decreased from 3240 L m^−2^ h^−1^ bar^−1^ at a Δ*P* of 1 kPa to 2016 L m^−2^ h^−1^ bar^−1^ at a Δ*P* of 6–10 kPa, corresponding to a 38% loss. For the second run of the pressure stepping filtration test, the initial clean water permeability decreased from 2823 L m^−2^ h^−1^ bar^−1^ at a Δ*P* of 1 kPa to 1676 L m^−2^ h^−1^ bar^−1^ at a Δ*P* of 10 kPa, corresponding to a 40% loss. For the first run, a small permeability fluctuation was observed for a Δ*P* of 6–10 kPa, while for the second run, a steady decline was observed at a Δ*P* of 1–10 kPa.

Significant changes in clean water permeability under the ultra-low-pressure range (less than 10 kPa) demonstrated the significant role of Δ*P* in dictating the hydraulic performance. This finding implies that data on clean water permeability from two independent studies obtained under different Δ*P*s can be significantly different. The same findings were obtained in our recent studies on ultra-low-pressure filtration. For a GDM filtration system and hollow fiber membrane, the clean water permeability decreased from 720, 500, to 426 L m^−2^ h^−1^ bar^−1^ when Δ*P* as increased from 2.2, 3.2, to 10.0 kPa, respectively [18]. Similarly, for a flat sheet membrane, the permeability decreased from 2740 to 376 L m^−2^ h^−1^ bar^−1^ when Δ*P* increased from 2.5 to 19.0 kPa, respectively [19].

When exposed to Δ*P*, the membrane structure is compressed by means of matrix deformation, pore constriction, and other phenomena. This phenomenon was reported in another study but for relatively high Δ*P*s. Applying Δ*P* during the filtration on a polyethersulfone hollow fiber membrane densified the porous support layer and thickened the skin layer (selective barrier) [24]. As a result, it also lowered the clean water permeability. Membrane compaction was also observed on the RO membrane, which reduced the water permeability and increased the support layer’s resistance, with more prominent effect at higher pressures and temperatures [25,26]. Membrane compaction also deformed the polymer matrix by lowering the membrane’s volume porosity [10] and altering the membrane pore size or pore geometry [10].

Offline examinations of membrane compaction (such as scanning electron microscopy analysis or micrometer measurements) lack accuracy in measuring its impact on filtration capacity. Various mechanical testing configurations were also attempted to assess membrane compaction [27,28]. However, the conditions used in these tests were much different from those found in a filter cell during filtering, leading to poor data accuracy.

Figure 2 also shows the instantaneous nature of membrane compaction under a small change in Δ*P* in the range of 1 to 10 kPa. For the first pressure stepping filtration run, the permeability was almost constant for Δ*P*s beyond 6 kPa, while a gradual decline was observed for the second run. The findings also show that membrane compaction is very sensitive to pressure under a low Δ*P* range (<6 kPa). Only minor changes in permeability were observed during the filtration (at a constant Δ*P*), indicating that the compaction instantaneously occurred when the filtration started and was poorly observed during the short-term filtration of 30 min. The slow compaction during the filtration is further discussed in Section 3.3.

Interestingly, a trend line could be generated to estimate the true intrinsic permeability of the membrane without any compaction (Figure 3), which is the permeability value where L vs. Δ*P* crosses the *y* axis. The polynomial trend line of the first run follows: L = 1.4567Δ*P*^4^ − 37.208Δ*P*^3^ + 346.3Δ*P*^2^ − 1428.9Δ*P* + 4350.7 with R^2^ = 0.9988, corresponding to an intrinsic permeability of 4350.7 L m^−2^ h^−1^ bar^−1^; that of the second run follows: L = 0.862Δ*P*^4^ − 23.448Δ*P*^3^ + 233.03Δ*P*^2^ − 1043Δ*P* + 3640.2 with R^2^ = 0.9959, corresponding to an intrinsic permeability of 3640.2 L m^−2^ h^−1^ bar^−1^. The permeability recovery at the start of the second run of filtration demonstrated that the compaction was highly reversible. The permeability at 1 kPa decreased by 13% from 3240 to 2823 L m^−2^ h^−1^ bar^−1^, while the decrease in the intrinsic permeability was 16%. The permeability values during the second run continued decreasing below the lowest value in the first run (1995 L m^−2^ h^−1^ bar^−1^), suggesting further compaction. Figure 3 differentiates the types of compactions in an ultra-low-pressure system comprising fast (instantaneous) and slow compactions. Fast compaction constitutes reversible and irreversible fractions.

### 3.2. Slow Compaction

Figure 4 shows the evolution of permeability over 8 h of constant filtration under three Δ*P*s of 1, 6, and 10 kPa. It shows no indication of slow compaction over the filtration duration, where the permeability changes from the initial to the final reading, ranging from −6% to 6%. When the two runs were averaged, the changes were between −1% and 3%. The negative range indicated that the final permeability was higher than the initial permeability due to a small variation in the experimental conditions. These results show that the phenomenon of membrane compaction under ultra-low-pressure membrane filtration instantaneously occurred when Δ*P* was applied to drive the permeation. This finding is consistent with the data obtained during a shorter filtration duration (30 min) shown in Figure 2.

Figure 4 also shows that the permeability for the second run of filtrations was lower than that for the first runs. This can be explained by the irreversible compaction obtained in the first run. Some degree of irreversible compaction was observed, even though the membrane was relaxed by letting it idle overnight (12 h). The compaction and decompaction phenomena resulted from filtration relaxation in the ultra-low-pressure filtration might complicate the analysis of the actual filtration when the feed is also prone to fouling the membrane. In this case, a confounding effect of membrane fouling and membrane compaction would be expected, even when the fouling rate is very severe, as observed earlier for the treatment of tapioca wastewater in a GDM bioreactor [18].

### 3.3. Compaction Dynamics

Figure 5 shows the relaxation effect on membrane compaction dynamics during six filtrations (P1–P6) under a Δ*P* of 6 kPa with relaxations. Each filtration was run for 30 min, followed by 60 min of relaxation. Figure 5A shows no evidence of irreversible compaction over the first five cycles (filtration and relaxation). Only a slight decrease (about 3%) in permeability was observed for the sixth cycle. However, the change was insignificant based on a statistical analysis using the post hoc Tukey HSD test with a Tukey HSD p-value of less than 0.9. This suggests that relaxation restored all reversible compactions from the previous cycles.

Figure 5B shows the initial and final values of the permeabilities for the six cycles. A slight decrease in permeability was observed during the 60 min filtration. However, this decrease was fully restored after the relaxation at the beginning of the next cycle. The ability of relaxation to restore the permeability was evident during the first four cycles and the first three cycles of the second run. Beyond those cycles, relaxation poorly restored permeability. The initial permeability of the final cycle of run 1 was 2074 L m^−2^ h^−1^ bar^−1^, while the initial permeability of the first cycle of run 2 was 2112 L m^−2^ h^−1^ bar^−1^, which was 1.8% higher. This was considered as part of the compaction dynamics due to the prolonged relaxation between the two runs. Nevertheless, the permeability decline was statistically insignificant. This finding is important when considering the operation of ultra-low-pressure filtration in a submerged MBR equipped with a plate-and-frame module with a flat-sheet membrane that typically operates under Δ*P* < 0.3 bar [29]. Traditionally, the permeability loss over time in an MBR operation has exclusively been attributed to membrane fouling [27,30]. However, the confounding effect of compaction and decompaction is expected, as ascribed from the findings in this study. In the past, the occurrence of compaction and decompaction was grouped into membrane fouling. Such limitations can hamper the judgment and mislead the analysis, leading to inaccurate results. For instance, during the filtration of activated sludge, the applied pressure was found to significantly affect the permeability in an ultra-low-pressure filtration system [18].

Figure 6 shows the filtration dynamics under two alternating Δ*P*s of 1 and 10 kPa. It clearly shows a slight decrease in permeability as the filtration proceeded. The permeability decline trend was slightly more prominent for filtrations under a Δ*P* of 1 kPa. Unlike the full relaxation shown in Figure 5, where permeability could fully be restored, holding the filtration at 1 kPa inhibited the complete restoration of the permeability. The permeability degeneration shown for a Δ*P* of 1 kPa filtration represents the irreversible compaction illustrated in Figure 3. The filtration test involving a very low Δ*P* was also used to identify the irreversible fouling rate in colloid filtration, mainly for MBR application [31].

The presence of irreversible compaction demonstrated in Figure 6 suggests the critical role of membrane compaction and decompaction in ultra-low-pressure membrane filtration. This role is expected to be highly dominant when dealing with feeds with a low membrane fouling propensity, involving periodic membrane cleanings to maintain high flux [32,33,34,35,36,37]. The trend in operating GDM under ultra-low pressure is increasing, and membrane compaction will play an essential role in such applications. In addition, ultra-low-pressure applications will benefit from developing custom-made membrane materials, as recently reported [38,39,40]. However, in addition to from combating membrane fouling, membrane material development should consider the compaction phenomenon that can substantially reduce permeability (>50%) from the intrinsic value, as shown in Figure 1.

## 4. Conclusions

This paper reports the phenomenon of membrane compaction in ultra-low filtration systems that have recently been used in emerging gravity-driven membrane (GDM) filtration of water and wastewater treatment. The findings demonstrated the prominent impact of membrane compaction under the tested pressures (Δ*P* < 10 kPa). Clean water permeability decreased from 3240 L m^−2^ h^−1^ bar^−1^ under a Δ*P* of 1 kPa to 2401 L m^−2^ h^−1^ bar^−1^ under a Δ*P* of 10 kPa (Figure 2). The actual permeability was found to be significantly lower than the intrinsic permeability (without compaction), with a permeability loss of >50% (4351 vs. 2401 L m^−2^ h^−1^ bar^−1^). The compaction was mainly instantaneous, with immediate permeability changes at different Δ*P*s. Only minor permeability changes were observed during prolonged clean water filtrations of 8 h under Δ*P*s of 1, 6, and 10 kPa. The membrane compaction was highly reversible and could be restored (i.e., decompaction) by applying relaxation via temporarily stopping the filtration. The permeability of six filtrations under a Δ*P* of 10 kPa was statistically insignificant. By alternatingly operating filtrations under a high (10 kPa) and low (1 kPa) Δ*P*, irreversible compaction was detected. Our findings are essential for supporting the emerging applications of GDMs in which the filtration is driven by the hydrostatic pressure at an ultra-low pressure of <10 kPa. In this context, the occurrence of compaction might be grouped into membrane fouling, resulting in an inaccurate analysis. Membrane compaction’s roles are expected to be more prominent in GDM filtration systems treating less-fouling-prone feed (i.e., rainwater or river water) and when involving membrane cleaning (i.e., relaxation). In this situation, the operational fluxes are relatively high; hence, both irreversible and irreversible compactions also occur. Further studies on the effect of compaction on solute rejection are strongly suggested. If compaction affects solute rejection, Δ*P* can be used as an important parameter to define permeate quality.

## Figures and Tables

**Figure 1 polymers-14-03254-f001:**
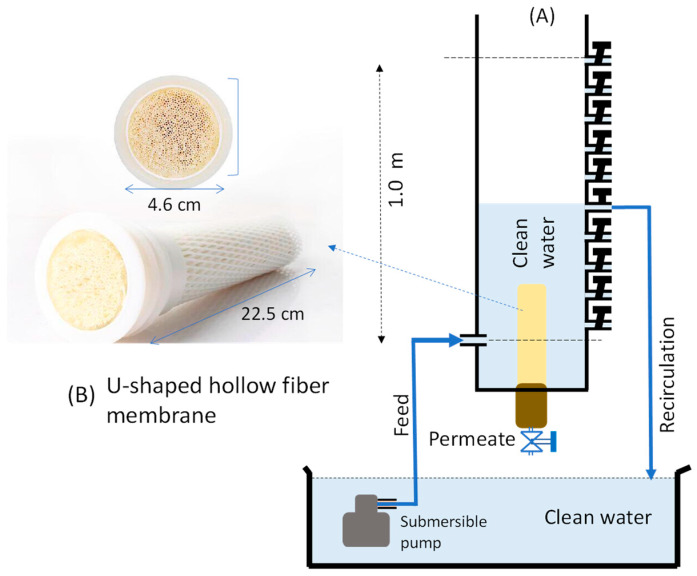
(**A**) A schematic illustration of a gravity-driven filtration setup used for clean water filtration and (**B**) pictures of the U-shaped hollow fiber membrane used for filtration.

**Figure 2 polymers-14-03254-f002:**
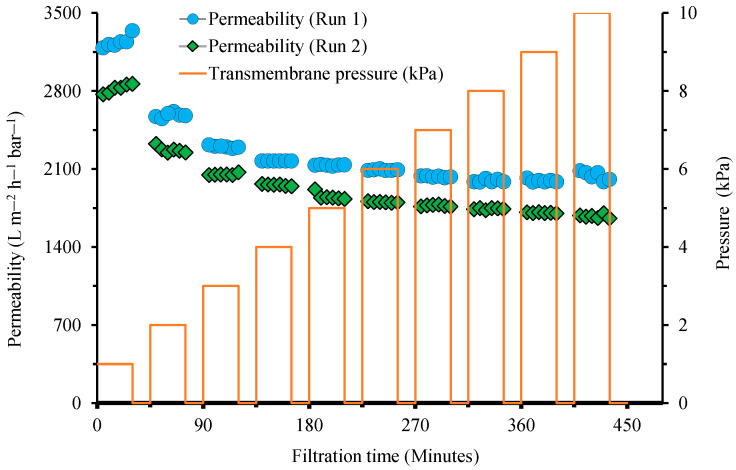
Evolution of clean water permeability during two runs of pressure stepping filtration test, showing the decreasing trend in permeability at higher transmembrane pressure.

**Figure 3 polymers-14-03254-f003:**
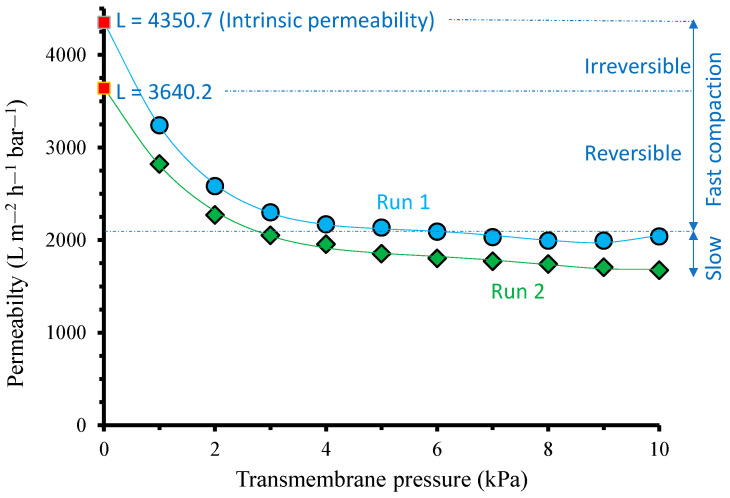
Intrinsic resistance of the membrane without experiencing compaction and decompaction rate. The polynomial trend line of the first run follows: L = 1.4567 Δ*P*^4^ − 37.208 Δ*P*^3^ + 346.3 Δ*P*^2^ − 1428.9 Δ*P* + 4350.7 with R^2^ = 0.9988, corresponding to an intrinsic permeability of 4350.7 L m^−2^ h^−1^ bar^−1^; that of the second run follows: L = 0.862 Δ*P*^4^ − 23.448 Δ*P*^3^ + 233.03 Δ*P*^2^ − 1043 Δ*P* + 3640.2 with R^2^ = 0.9959, corresponding to an intrinsic permeability of 3640.2 L m^−2^ h^−1^ bar^−1^.

**Figure 4 polymers-14-03254-f004:**
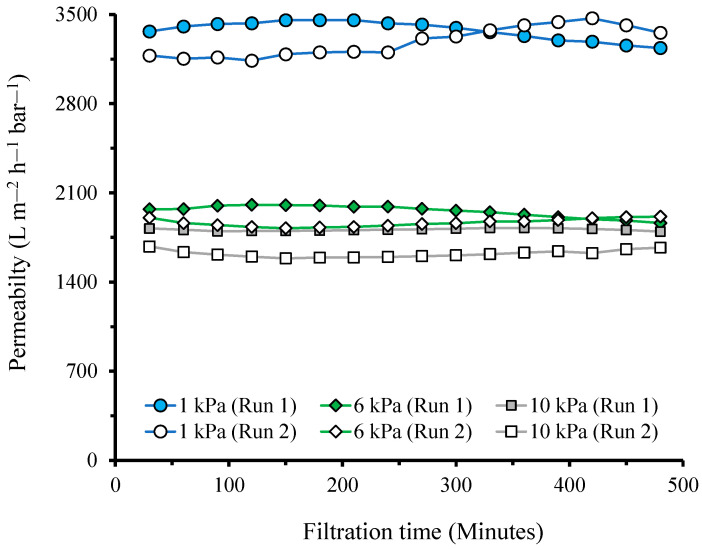
Evolution of permeability over an extended filtration time under various transmembrane pressures.

**Figure 5 polymers-14-03254-f005:**
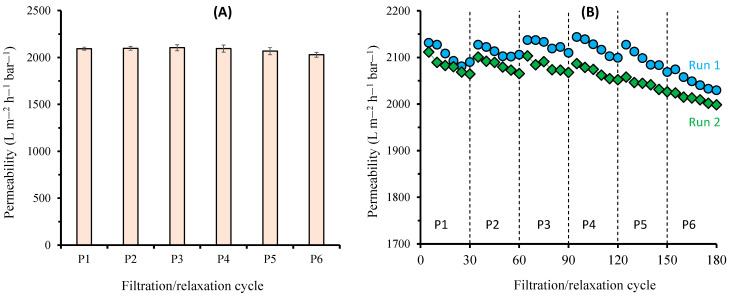
Compaction dynamics of the membrane under multiple filtrations (at constant pressure of 6 kPa) and relaxation cycles, showing (**A**) average permeability from two runs and (**B**) initial and final permeabilities of each cycle. Dashed lines in (**B**) represent relaxation/idle periods in between two runs for 60 min.

**Figure 6 polymers-14-03254-f006:**
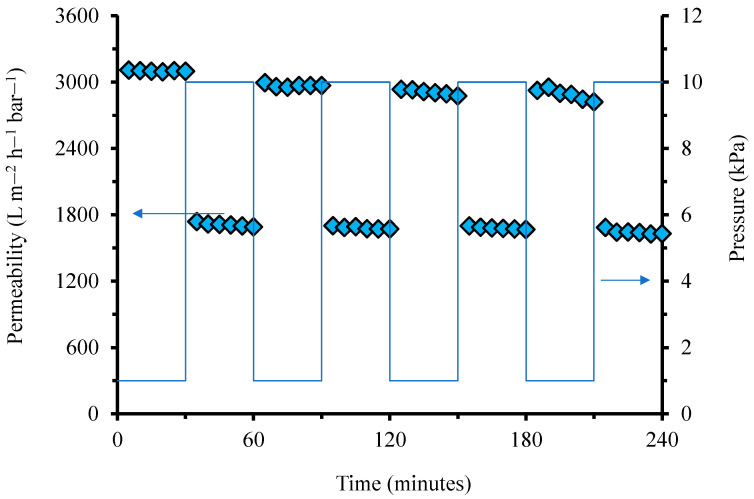
Evolution of permeability under two alternating transmembrane pressures of 1 and 10 kPa.

## Data Availability

Not applicable.

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
