# Peer review of "Compaction of a Polymeric Membrane in Ultra-Low-Pressure Water Filtration"

_polymers, 2022, doi:10.3390/polym14163254_

Round 1

Reviewer 1 Report

The manuscript reports the verification of the compaction of polymeric membrane through the study on how the filtration pressure affects the permeability under an ultra-low-pressure system. Some new findings are presented, and the manuscript is well written. I recommend its publication in Polymers. One suggestion to the authors could be that the study on the effect of the membrane compaction on solute rejection efficiency can be further studied.

Reviewer 2 Report

This is a very well written manuscript which describes the study of compaction on hollow fiber membranes in a gravity-driven filtration. The introduction part is particularly well written with clear summary of literature, problem statement and the current study. Overall, it is a clearly articulated report which is easy to understand. Importantly, it adds value to the scientific body of knowledge in the area of membrane science because the understanding the compaction issues is still not thorough not just GDM but also in high pressure applications such as NF and RO. As authors have pointed out, the compaction is misunderstood for fouling in some instances and this study certainly opens up the path for relooking the operational issues such as increase in transmembrane pressure and also whether the cleaning cycles need to be replaced with de-compaction to restore the flux. 

I recommend that this manuscript be accepted after some minor changes are made, as suggested below. 

1. Line 42, "reversible and irreversible"

2. Line 74 "matrix"

3. Section 2.1 Please describe the hollow fiber preparation method and some basic characterization such as MWCO, etc., if possible. 

4.  Please label the different sub-figures in Figure 1 as (a), (b), etc for clarity and also modify the caption accordingly. The U-shaped hollow fiber needs to be explained in text or through a figure, to make it easy for understanding. U-shaped means the HF is bent inside the module, with both ends seen in the module, which is the typical packing configuration for a dead-end filtration. 

5. Line 142. Please specify the intervals. is it in 1 kPa interval?

Reviewer 3 Report

The manuscript describes the study of the phenomenon known from the literature i.e. phenomenon of membrane structure compression when increasing the trans membrane pressure (TMP). The authors chose pure water as the working medium to investigate this phenomenon without the participation and influence of fouling on the reduction of membrane permeability. The authors have already described this phenomenon in their published works.

In this paper there is no description of the membrane module, which is the subject of the research.

The following papers by the same team used the same "U-shaped hollow fiber membrane" system. This can be seen on photos in the experimental setup of this and the below-mentioned works:

- Membranes 2021, 11 (11), 875; https://doi.org/10.3390/membranes11110875 - Sequencing Batch Integrated Fixed-Film Activated Sludge Membrane Process for Treatment of Tapioca Processing Wastewater, by Nur Izzati Zainuddin, Muhammad Roil Bilad, Lisendra Marbelia, Wiratni Budhijanto, Nasrul Arahman, Afrilia Fahrina, Norazanita Shamsuddin, Zaki Ismail Zaki, Zeinhom M. El-Bahy, Asep Bayu Dani Nandiyanto and Poernomo Gunawan

- Membranes 2022, 12, 591. https://doi.org/10.3390/membranes12060591

Ultra-Low-Pressure Membrane Filtration for Simultaneous Recovery of Detergent and Water from LaundryWastewater. by Khery, Y .; Daniar, S.E .; Mat Nawi, N.I .; Bilad, M.R.; Wibisono, Y .; Nufid, B.A .; Ahmadi, A .; Jaafar, J .; Huda, N .; Kobun, R.

The characteristics of the membranes used were not given in any of these works. An exemplary,  correct description of another membrane can be found in [22] listed in References. It contains the manufacturer's data, brand name, membrane material, membrane configuration, MWCO, nominal permeability, wall thickness, inner diameter and module membrane area.

If the authors do not provide such data, their manuscript should be rejected as not meeting the basic condition of scientific work - the possibility of replication of research and confirmation of the results.

 Some specific comments:

1) I believe that the description of the polymer membrane used in the process is too sketchy. There is no reliable information about the polymer material (PAN) from which the membrane was made - the method of its preparation, manufacturer and purity - which the nature of the Polymers journal would require.

2) Filter characteristics are not given - manufacturer, symbol, hydrodynamic operating parameters? It does not appear from the manuscript that this has been described in previous studies.

3) Figure 1: what was the diameter of the tank or maximum volume? U-shape hollow fiber membrane (filtration cartridge) should be better described - average pore diameter, membrane thickness, material of the membrane, etc.

4) Methodology: hollow fiber membrane module is vertical and has height ca, 22 cm, so on the top of this module pressure is 2 kPa higher than on the lower part. The question is, to which point of the module do they refer by writing 1, 6 and 10 kPa? Is it the middle of the module height?

5) The language used in manuscript requires improvement. A number of stylistic errors can be noted:

In title "liquid" should be replaced by "water".

Line 192 - word "matric" should be "matrix"

Line 198 - remove word "coefficient"

Line 232 word "and" is doubled

Figures 3 and 4 use "Round 1" and "Run 1", is it the same?

In the caption to Figure 5, it should be clearly stated that it concerns a pressure of 6 kPa. If Fig 6A and 6B are related to the same experiment then the data on the "Permeability" axis does not agree. 2000 vs 2400-2600.

Figure 6. The data on the diagram shows that the Flux for a pressure of 10 kPa will be greater than five times the Flux for a pressure of 1 kPa. Thus, under industrial conditions, higher operating pressures for the membrane will be selected. Authors should make appropriate graphs to show Flux for pressures. The only factor that may prompt the use of lower TMPs is the lack of fouling in subcritical cross-flow condition.

Line 314-317 The number 3240 is read from Fig 6, but in this figure no data is near 2401. It is given around 1700. The only place where this number can occur is in Fig 5B - the last point. However, it seems the graph is made for a TMP of 6 kPa, not 10 kPa

Final conclusion:

In its current form, the manuscript is not suitable for publication as it does not have the main feature of a scientific publication - the possibility of repeating and verifying the results (no descriptions of the membrane polymeric material). It presents too little scope of research.

Reviewer 4 Report

The manuscript is well organized on the membrane compaction at low pressure, however, it needs to be modified referring to the following.

1. Membrane compaction is thought to be affected by fabrication method, material and porosity of the membrane. So it would be good to add related contents or references to the introduction.

2. Compaction fraction (reversible vs irreversible and slow vs fast) is shown via two consecutive runs using the same membrane in each experiment. It would be meaningful to show how the membrane compaction proceeds in the case of continuous operation more than 3 times.

3. Graph B seems to be strange in Figure 5. Run 2 would be carried out using the same membrane which is compacted irreversibly at the end of run 1, but the initial permeability of the first cycle of run 2 is higher than the one of t the last cycle of run 1.

Round 2

Reviewer 3 Report

The manuscript describes the study of the phenomenon known from the literature i.e. phenomenon of membrane structure compression when increasing the trans membrane pressure (TMP). The authors chose pure water as the working medium to investigate this phenomenon without the participation and influence of fouling on the reduction of membrane permeability. The authors have already described this phenomenon in their published works.

A number of corrections were introduced in the work, which resulted from the reviewers' comments, but the main defects of the work were not removed. The reviewers asked for more detailed characteristics of the material of the membrane module. The authors did not provide this data, and as the manufacturer they wrote "OEM, China" which means an unknown manufacturer.

 In this paper there is no detailed description of the membrane module, which is the subject of the research.

An exemplary, correct description of another membrane can be found in [24] listed in References. It contains the manufacturer's data, brand name, membrane material, membrane configuration, MWCO, nominal permeability, wall thickness, inner diameter and module membrane area. Standard works in this field generally include SEM photographs of the surface and cross-section of the membrane.

If the authors do not provide such data, their manuscript may be rejected as not meeting the basic condition of scientific work - the possibility of replication of research and confirmation of the results.

 Some specific comments:

1) I believe that the description of the polymer membrane used in the process is too sketchy. There is no reliable information about the polymer material (PAN) from which the membrane was made - the method of its preparation, name of the manufacturer and purity of the polymer - which the nature of the "Polymers" journal would require.

2) Filter characteristics are not given - manufacturer, symbol, hydrodynamic operating parameters?

Figures 3 and 4 use "Round 1" and "Run 1", is it the same? Fig 3 was not corrected.

3) Authors should carefully check their corrections. For example, in the corrected caption to Figure 5, there is one typing error!

4) The authors increased the number of References to 42. However, one of the References [10] is already used as [27]. Therefore, most references in the text should be renumbered.

Final conclusion:

In its current form, the manuscript is not suitable for publication as it does not have the main feature of a scientific publication - the possibility of repeating and verifying the results (no descriptions of the membrane polymeric material).

In my opinion, the authors should repeat the experiments for a membrane module without an outer sheath that keeps the U-tubes close together and may cause the tubes to stick together and thus partially mechanically block the filter surface. Increasing pressure can deform the shape of the tubes from circular to hexagonal and thus reduce the free filtration area. The outer sheath of the module may favor the hydrodynamics of this phenomenon. Please note that in most published case studies, U-shaped hollow fiber-tube modules float freely.
